# Protein energy wasting in pre-dialysis chronic kidney disease patients in Benin City, Nigeria: A cross-sectional study

Osariemen Augustine Osunbor[1], Evelyn Irobere Unuigbe[2], Enajite Ibiene Okaka[3], Oluseyi Ademola Adejumo[4]*

1 Department of Internal Medicine, Central Hospital, Benin City, Edo State, Nigeria, 2 Moonshine Elder Care Centre, Benin City, Edo State, Nigeria, 3 Department of Internal Medicine, University of Benin Teaching Hospital, Benin City, Edo State, Nigeria, 4 Department of Internal Medicine, University of Medical Sciences, Ondo City, Ondo State, Nigeria

* oluseyiadejumo2017@gmail.com

## Abstract

### Introduction

Protein energy wasting (PEW) is common among chronic kidney disease (CKD) patients, especially those with advanced stage. It worsens frailty, sarcopenia and debility in CKD patients. Despite the importance of PEW, it is not routinely assessed during management of CKD patients in Nigeria. The prevalence of PEW and its associated factors were determined in pre-dialysis CKD patients.

### Methods

This was a cross-sectional study that involved 250 pre-dialysis CKD patients and 125 age- and sex- matched healthy controls. Body mass index (BMI), subjective global assessment (SGA) scores and serum albumin levels were used in PEW assessment. The factors associated with PEW were identified. P-value of < 0.05 was taken as significant.

### Results

The mean age of CKD and control group were 52.3±16.0 years and 50.5±16.0 years, respectively. The prevalence of low BMI, hypoalbuminaemia and malnutrition defined by SGA in pre-dialysis CKD patients were 42.4%, 62.0% and 74.8%, respectively. The overall prevalence of PEW among the pre-dialysis CKD patients was 33.3%. On multiple logistic regression, the factors associated with PEW in CKD were being middle aged (adjusted odds ratio: 12.50; confidence interval: 3.42–45.00; p <0.001), depression (adjusted odds ratio: 2.34; confidence interval: 1.02–5.40; p = 0.046) and CKD stage 5 (adjusted odds ratio: 12.83; confidence interval: 3.53–46.60; p <0.001).

### Conclusion

PEW is common in pre-dialysis CKD patients and it was associated with middle age, depression and advanced CKD. Early intervention aimed at addressing

**Data Availability Statement:** All relevant data are within the paper and its Supporting Information files.

**Funding:** NO- The funders had no role in study design, data collection and analysis, decision to publish, or preparation of the manuscript.

**Competing interests:** The authors have declared that no competing interests exist.

depression in early stages of CKD may prevent PEW and improve overall outcome in CKD patients.

## Introduction

Protein energy wasting (PEW) is common among chronic kidney disease (CKD) patients, but often not diagnosed early in Nigeria and it worsens with progressive CKD. It is characterized by loss of muscle and fat arising from concurrent loss of body protein and energy stores [1]. It involves the combination of both nutritional and metabolic alteration in the CKD patients.

The factors responsible for PEW in CKD patients are multifactorial and include chronic systemic inflammation, metabolic acidosis, oxidative stress, intestinal dysbiosis, hormonal disorders, reduced nutritional intake, increased energy expenditure, and uraemic toxins [2–5]. Chronic inflammation is common in CKD patients on dialysis and this leads to PEW with increased protein catabolism [6]. PEW reduces functional capacity and worsens frailty, sarcopenia and debility in CKD patients [1, 5, 6].

According to some previous studies, the prevalence of PEW in CKD patients range between 23.4–74.5% [7–14]. The differences in socio-economic characteristics of CKD patients, study methodology and criteria used in the diagnosis of PEW partly account for the wide differences in reported prevalence rates from previous studies. The prevalence of PEW is higher with more advanced CKD and in patients on dialysis [7–14].

There are only few studies on assessment of nutritional status in pre-dialysis CKD patients in Sub-Saharan Africa including Nigeria [11, 15–17]. This study aimed to give an overall impression of the scope of PEW in CKD in Nigeria. This will increase awareness and facilitate prompt diagnosis and treatment in high-risk individuals.

## Materials and methods

This was a cross-sectional study carried out at the University of Benin Teaching Hospital (UBTH), Benin City, Edo State, Nigeria. This study was carried out over a fourteen-month period from October 2016 to November 2017. Dialysis naive CKD patients attending the nephrology consultant out-patient clinic in UBTH who met the inclusion criteria were consecutively recruited for the study.

The sample size for this study was determined using Fleiss formula. The prevalence of PEW in CKD patients and the general population used in this calculation were 47.3% and 21.3%, respectively [9]. The confidence interval was taken as 95% and the power of study was 80%. The minimum sample size for this study was 80 and 40 for CKD and control groups respectively, using ratio 2:1 after including 10% attrition.

The inclusion criteria were: CKD patients aged 18 years and above, patients with stage 3–5 CKD and those who gave informed consent. Those who were on maintenance haemodialysis, steroids and kidney transplant patients were excluded. Consenting apparently healthy individuals who did not have CKD or other chronic medical conditions were recruited from the workforce of the hospital as control participants. Two-hundred and fifty pre-dialysis CKD patients and one hundred and twenty-five age and sex-matched apparently healthy controls without CKD were recruited for this study.

Study participants were interviewed using an investigator-administered structured questionnaire. Socio-demographic information such as age, sex, marital status, occupation, educational status and medical history were obtained. They went through physical examination

which included weight and height. Body mass index (BMI) was calculated from values of weight expressed in kg divided by the square of height in meters. Aetiology of CKD was determined based on the nephrologist's opinion using clinical history, physical examination and investigation results. Ten ml of fasting venous blood sample was collected for laboratory investigations which included fasting blood glucose, full blood count, erythrocyte sedimentation rate, serum total cholesterol, urea, creatinine, electrolytes and albumin.

The 7-point subjective global assessment (SGA) tool was used to assess malnutrition in the study population [18]. The SGA questionnaire has 3 parts; the first part is based on medical history, a review of the medical history including assessment of weight and weight changes, dietary intake, gastrointestinal symptoms, disease state/co-morbidity and the subject's functional capacity related to nutritional status; the second part is physical examination which focuses on loss of subcutaneous fat, presence of ankle or pedal oedema and ascites related to nutritional status; and the third part is the SGA scoring as A (well nourished), B (mildly-moderately malnourished and C (severely malnourished). Any participant with scores in the B or C categories was considered to be malnourished. Assessment of anorexia was based on SGA findings of self-reported intake and loss of appetite.

Modification of Diet in Renal Disease (MDRD) formula was used to estimate the glomerular filtration rate and determine the stage of CKD [19].

Depression was assessed using Hamilton Depression rating scale which has been validated and reported to be reliable in CKD population [20, 21]. Eight items are scored on a 5-point scale, ranging from 0 = not present, to 4 = severe. Nine items are scored from 0–2. Although the Hamilton Depression form lists 21 items, the scoring is based on the first 17. A score of zero to 7 is considered as normal, 8–13 mild depression, 14–18 moderate depression, 19–22 severe depression and $\geq$23 very severe depression.

PEW in this study was diagnosed in patients using a combination of at least two of the following indices:

i.  BMI $<23$kg/m$^2$ [1]

ii.  Serum albumin $< 3.8$g/dl [1]

iii.  Overall SGA score of B or C [18]

## Ethical consideration

Approval was obtained from the Ethical Committee of UBTH before commencement of the study. The protocol approval reference number was ADM/22/A/Vol.11/1269. All participants involved in this study gave written informed consent. All information about the study participants were treated with utmost confidentiality.

## Data analysis

Data entry and analysis were done using the IBM SPSS (Statistical Package for Social Sciences) version 21.0 software. Discrete variables were presented as frequencies and percentages. Univariate analysis was used in description of socio-demographic characteristics of the study participants. Continuous data were presented as means and standard deviation and student's t-test was used to compare means. Bivariate analysis was used to assess association between PEW and socio-demographic variables. Logistic regression was used to assess significant predictors of PEW after adjusting for other variables. Statistical significance was taken at p-value $< 0.05$.

## Results

A total of 375 participants were studied comprising of 250 CKD patients and 125 healthy controls. The CKD and control groups were well matched for age and gender. The CKD group was made up of 116(46.4%) males and 134(53.6%) females. The mean age of the CKD and control groups were 52.3±16.0 years and 50.6±16.1 years, respectively. The common causes of CKD among participants were hypertension in 83 (33.2%), chronic glomerulonephritis in 67 (26.8%) and diabetes mellitus in 63 (26.0%) Table 1.

The mean serum urea, creatinine, potassium, and erythrocyte sedimentation rate (ESR) were significantly higher in the CKD group compared to the control group (p<0.001). Haematocrit, serum albumin and estimated GFR were significantly lower in the CKD group compared to the control group (0<0.001) Table 2.

**Table 1. Socio-demographic and clinical characteristics of study participants (N = 375).**

| Characteristic | CKD Group (n = 250) Frequency (%) | Control Group (n = 125) Frequency (%) | P-value |
|---|---|---|---|
| **Sex** | | | |
| Male | 116 (46.4) | 59 (47.2) | 0.884 |
| Female | 134 (53.6) | 66 (52.8) | |
| **Mean Age (years)** | 52.3±16.0 | 50.6±16.1 | |
| **Age group (years)** | | | |
| <30 | 20 (8.0) | 15 (12.8) | 0.333 |
| 30–49 | 84 (33.6) | 43 (34.4) | |
| 50–69 | 110 (44.0) | 51 (40.8) | |
| ≥70 | 36 (14.4) | 16 (12.8) | |
| **Marital status** | | | |
| Married | 203(81.2) | 58(46.4) | |
| Not married | 47(18.8) | 67(53.6) | |
| **Educational status** | | | |
| Below tertiary | 137 (54.8) | 22(17.6) | |
| Tertiary | 113 (45.2) | 103(82.4) | |
| **Employment status** | | | |
| Employed | 197(78.8) | 119(95.2) | |
| Unemployed | 53(21.2) | 6(4.8) | |
| **CKD Stage** | | | |
| 3 | 135(54.0) | | |
| 4 | 89(35.6) | | |
| 5 | 26(10.4) | | |
| **Aetiology of CKD** | | | |
| Hypertension | 83(33.2) | | |
| Chronic Glomerulonephritis | 67(26.8) | | |
| Diabetes Mellitus | 63(26.0) | | |
| Human Immunodeficiency virus | 20(14.0) | | |
| Sickle Cell Disease | 11(4.4) | | |
| Obstructive uropathy | 4(1.6) | | |
| **Duration of CKD** | | | |
| <3 months | 111(44.4) | | |
| 3–6 months | 47(18.8) | | |
| 7–12 months | 33(13.2) | | |
| >12 months | 59(23.6) | | |

**Table 2. Biochemical and haematological parameters of study participants (N = 375).**

| Parameter | CKD Group (n = 250) Mean ±SD | Control Group (n = 125) Mean±SD | P-value |
|---|---|---|---|
| Sodium (mmol/L) | 134.5 ±7.2 | 137.3±4.2 | <0.001 |
| Potassium (mmol/L) | 4.4±0.9 | 3.8±0.3 | <0.001 |
| Bicarbonate (mmol/L) | 19.1±3.9 | 21.0±1.2 | <0.001 |
| Chloride (mmol/L) | 102.0±5.8 | 102.2±3.4 | 0.732 |
| Urea (mg/dl) | 108.3±49.8 | 34.5±11.6 | <0.001 |
| Creatinine (mg/dl) | 2.8±1.9 | 0.8± 0.2 | <0.001 |
| Calcium total (mg/dl) | 8.1±1.1 | 8.9±0.4 | <0.001 |
| Phosphate (mg/dl) | 4.0 ±0.9 | 4.9±(5.9 | 0.022 |
| eGFR (ml/min) | 30.0±12.2 | 103.9±27.7 | <0.001 |
| FBG (mg/dl) | 110.7±60.9 | 84.5±10.9 | <0.001 |
| Packed cell volume (%) | 27.3±7.1 | 40.1±6.6 | <0.001 |
| Haemoglobin (g/%) | 9.4±2.5 | 14.7±10.6 | <0.001 |
| ESR (mm/h) | 76.4±27.1 | 20.3±13.5 | <0.001 |
| Serum albumin(g/dl) | 3.2±0.9 | 4.0±0.3 | <0.001 |
| Serum TC(mg/dl) | 170.3±53.6 | 173.4±30.6 | 0.477 |

FBG: fasting blood glucose, TC: total cholesterol, ESR: erythrocyte sedimentation rate, eGFR: estimated glomerular filtration rate

The prevalence of low BMI, low serum albumin and malnutrition based on SGA were 42.4%, 62.0% and 74.8%, respectively. The overall prevalence of PEW using the combination of these criteria in the pre-dialysis CKD patients was 33.3% while none of the participants in the control group had PEW Table 3.

On bivariate analysis, there was significant association between PEW in the CKD patients and age (p < 0.001), income (p = 0.001), marital status (p = 0.022), CKD duration (p = 0.001), CKD stage (p < 0.001), anorexia (p < 0.001), depression (p = 0.002), and aetiology of CKD Table 4.

On multiple logistic regression, the factors associated with PEW in CKD were being middle aged (adjusted odds ratio: 12.50; confidence interval: 3.42–45.00; p <0.001), depression (adjusted odds ratio: 2.34; confidence interval: 1.02–5.40; p = 0.046) and advanced CKD stage (adjusted odds ratio: 12.83; confidence interval: 3.53–46.60; p <0.001) Table 5.

## Discussion

This study aimed to give an overall impression of the scope of PEW in a moderate-severe out-patient CKD cohort in Nigeria in order to raise awareness of the problem and encourage early intervention. The prevalence of low BMI, low serum albumin and malnutrition based on SGA were 42.4%, 62.0% and 74.8%, respectively. The overall prevalence of PEW using the

**Table 3. Prevalence of malnutrition indices among study participants (N = 375).**

| Malnutrition Index | CKD Group (n = 250) Frequency (%) | Control Group (n = 125) Frequency (%) | P-value |
|---|---|---|---|
| Low Body Mass Index | 106 (42.4) | 28 (22.4) | <0.001 |
| Low Serum Albumin | 155 (62.0) | 29(23.2) | <0.001 |
| SGA | 187 (74.8) | 0 (0.0) | <0.001 |
| PEW | 83 (33.2) | 0(0.0) | <0.001 |

SGA: Subjective Global Assessment PEW: Protein energy wasting

**Table 4. Factors associated with PEW in CKD patients (N = 250).**

| Variables | Cases | | P-value |
| --- | --- | --- | --- |
| | Malnourished | Not malnourished | |
| | n = 83 | n = 167 | |
| | Frequency (%) | Frequency (%) | |
| **Gender** | | | |
| Male | 35 (30.2) | 81(69.8) | 0.209 |
| Female | 48 (35.8) | 86 (64.2) | |
| **Age group (years)** | | | |
| <45 | 45 (56.3) | 35 (43.7) | <0.001 |
| 45–64 | 29 (27.4) | 77 (72.6) | |
| ≥65 | 9 (14.1) | 55(75.9) | |
| **Income** | | | |
| <18,500 naira | 8(47.1) | 9(52.9) | |
| 18,500–85,000 naira | 18(72.0) | 7(28.0) | 0.001 |
| >85,000 | 57(27.4) | 151(72.6) | |
| **Marital status** | | | |
| Not married | 22(46.8) | 25(53.2) | 0.022 |
| Married | 61 (30.0) | 142(70.0) | |
| **Educational status** | | | |
| Below tertiary | 52 (32.0) | 85(68.0) | 0.052 |
| Tertiary | 31 (27.4) | 82(62.3) | |
| **Employment status** | | | |
| Employed | 63 (78.7) | 42 (21.3) | 0.617 |
| Unemployed | 20 (75.5) | 13 (24.5) | |
| **CKD Duration** | | | |
| < 3months | 51(45.9) | 60(54.1) | 0.001 |
| 3–6 months | 11(23.4) | 36(76.6) | |
| 7–12 months | 9(27.3) | 24(72.7) | |
| >12 months | 12(20.3) | 47(79.7) | |
| **CKD stage** | | | |
| Stage 3 | 24 (17.8) | 111 (82.2) | <0.001 |
| Stage 4 | 39 (43.8) | 50 (56.2) | |
| Stage 5 | 20 (76.9) | 6 (23.1) | |
| **Aetiology of CKD** | | | |
| Hypertension | 68 (81.9) | 15 (18.1) | |
| Chronic Glomerulonephritis | 55(82.1) | 12 (17.9) | |
| Diabetes Mellitus | 43 (66.2) | 22 (33.8) | 0.09 |
| HIV | 18(90.0) | 2(10.0) | |
| Sickle cell disease | 10(90.9) | 1(9.1) | |
| Obstructive uropathy | 3(75.0) | 1(25.0) | |
| **Anorexia** | | | |
| Present | 58(40.9) | 84 (59.2) | 0.002 |
| Absent | 25(23.1) | 83 (76.9) | |
| **Depression** | | | |
| Present | 28(50.0) | 28 (50.0) | 0.002 |
| Absent | 55 (28.4) | 139(71.6) | |

**Table 5. Predictors of PEW among the CKD patients.**

| Predictor | AOR (95% CI) | p-value |
|---|---|---|
| Gender | 0.75 (0.35–1.61) | 0.458 |
| Age (years) | | |
| <45 | 1 | |
| 44–64 | 12.47(3.42–45.0) | <0.001 |
| ≥65 | 5.56(1.0–18.16) | 0.004 |
| Marital status | 0.77 (0.42–3.18) | 0.462 |
| Educational level | 0.66(0.32–1.37) | 0.265 |
| Income level | 0.45(0.11–1.95) | 0.287 |
| Employment status | '1.27(0.47–3.19) | 0.639 |
| Anorexia | 1.32(0.62–2.81) | 0.479 |
| Depression | 2.34(1.02–5.40) | 0.046 |
| Duration of CKD | 1.15(0.83–1.58) | 0.403 |
| Stage of CKD | | |
| Stage 3 | 1 | |
| Stage 4 | 3.50(1.62–7.56) | 0.001 |
| Stage 5 | 12.83(3.53–46.60) | <0.001 |

combination of these criteria in the pre-dialysis CKD patients was 33.3% while none of the participants in the control had PEW.

The prevalence rates of PEW in various studies may vary due to differences in methodology, especially in the criteria used for diagnosis, region of study, participants' social, demographic and economic characteristics; hence this should be considered when comparing results The prevalence of PEW in this study is similar to 31.4% reported in a previous study in Nigeria [16]. It is also comparable with the pooled prevalence of 38.5% reported in India, a country with similar socio-economic characteristics with Nigeria where the present study was done [11]. However, it is higher than 23.4% reported in a study done in Ethiopia [8]. Although, the two studies were done in Africa with similar socio-economic characteristics, this difference may be partly accounted for by the criteria used in the diagnosis of PEW which were not the same. For instance, low BMI was defined as a value less than $18.5kg/m^2$ in the study done in Ethiopia unlike the present study where less than $23kg/m^2$ was used as recommended by the International Society of Renal Nutrition and Metabolism.

Similarly, the prevalence of low BMI in this study was 42.4% which is higher than 21.6% reported in a previous study done in Nigeria where BMI value of below $20kg/m^2$ was used [12]. However, it is lower than 62.4% reported in a study done in Australia [22]. The variation in the cut off values used to define low BMI and the age of participants in this study may account for this difference. The present study comprised of younger patients who were more likely to have relatively higher BMI due to their muscle and bone mass compared to older CKD cohorts in the study done in Australia. Although BMI is easy to assess, it is not sufficient as a nutritional parameter when used in isolation because factors such as fluid retention and race may affect it [1]. Low BMI is associated with higher morbidity and mortality in CKD patients [23]. In addition, a systematic review and meta-analysis by Ladhani et al. [24] showed that the risk of death in CKD stages 3–5 is reduced by 1% for every $1kg/m^2$ increase in BMI.

The prevalence of hypoalbuminaemia was 62.0% in this study. This is higher than 43.2% reported by Agaba et al. [12] among pre-dialysis CKD patients in a previous study in Nigeria. This difference may also be partly due to a lower cut off value of less than 2.9g/dl used to define hypoalbuminaemia unlike the present study where less than 3.8g/dl was used. The use of

serum albumin as a nutrition marker also has some limitations because it can be affected by plasma volume, inflammation, infections and dialysis in those who are on renal replacement therapy [25].

The prevalence of malnutrition using SGA as a tool for assessment in this study was 74.8%. The SGA is a simple and inexpensive questionnaire based tool that is highly reliable in assessing nutritional state in CKD [18]. It also shows significant association with various markers of nutrition such as anthropometric measurements and serum albumin [9]. The prevalence is higher than 28.6% and 40.5% reported in studies done in Italy and Australia which are high-resource countries compared to Nigeria where the present study was done [22, 26]. The fact that a large proportion of the CKD patients in the present study were newly diagnosed and therefore were probably not medically optimized may partly account for this difference. In addition, the patients' low economic status may also be a contributing factor. The prevalence of malnutrition using SGA in this study is also higher than 65% and 46% reported in previous studies conducted in India and Nigeria, respectively [13, 16]. However, the prevalence reported in this study is lower than 85.7% reported in a study done in China that involved older patients compared to the present study [10].

There was significant association between PEW and some socio-demographic factors in this study. PEW was associated with lower income which is similar to a report by Anupama et al. [27]. This may be because those with lower income are likely to have limited financial capacity to adequately feed well. Younger age and being unmarried were also associated with PEW which is similar to the report by Namuyimbwa et al. [7]. However, some other studies reported significant association between older age and PEW in their CKD population [8, 10]. PEW was also significantly more common in recently diagnosed CKD patients. This may be because they have not had ample time to be adequately optimized with treatment.

This study showed that CKD patients with depression were 2.3 times more likely to have PEW compared to those without depression. This is similar to reports from previous studies that showed significant association between depression and PEW in CKD patients [28, 29]. Depression is associated with activation of pro-inflammatory cytokines that may contribute to PEW and anorexia [30]. In addition, poor appetite and food intake that characterize depression may also lead to malnutrition in CKD. Hence, it is imperative that routine mental health assessment be done for patients suffering from CKD. Early diagnosis and treatment of depression may prevent PEW and improve overall outcomes in CKD patients.

There was significant association between stage of CKD and PEW in this study which is similar to findings of some previous studies [8, 31, 32]. The prevalence of PEW among CKD patients in stage 3, 4, 5 were 17.8%, 43.1% and 76.1%, respectively. Patients with CKD stage 4 and 5 were 3.5 times and 12.8 times more likely to have PEW compared to those with CKD stage 3, respectively in this study. This finding underscores the need for prompt evaluation, diagnosis and management of PEW during early stages of CKD. The worsening nutritional status with the progression of renal impairment observed could be associated with reduced or inadequate dietary intake due to anorexia and vomiting caused by ureamia, increased protein catabolism in skeletal muscles and chronic inflammation which worsen with declining GFR.

Anorexia was found to be significantly associated with PEW in this study which is similar to some previous reports [8]. Although, anorexia was subjectively assessed in this study, objective tools such as Visual Analogue Scale (VAS) and Functional Assessment of Anorexia/Cachexia Therapy (FAACT) have been validated and found useful in CKD populations [33]. There are changes in the level of hormones and inflammatory cytokines that regulate appetite in CKD leading to anorexia [4, 34]. Sahathevan et al. [34] reported low levels of acyl-ghrelin and increased levels of leptin, tumour necrosis factor-alpha and interleukin-6 in CKD patients which may suppress appetite and consequently lead to poor nutritional status in CKD patients.

Some measures that have been effective in the management of PEW in CKD patients include dietary counselling, optimizing dietary intake, physical exercise, oral nutritional supplementation (ONS), and correction of metabolic disturbances such as metabolic acidosis, hormonal deficiencies and systemic inflammation [35, 36]. Vijaya et al. [37] reported significant improvement of nutritional parameters in CKD patients following dietary counselling and intervention by dieticians. Wong et al. [38] also reported significant increase in some nutritional parameters such as BMI and serum albumin and reduction in inflammatory markers with ONS. Nutritional status and general well-being of CKD patients could be improved by regular physical exercise which strengthens muscle and prevents muscle breakdown [36]. Early dietary counselling, optimization of dietary intake, correction of metabolic acidosis and regular physical exercise are sustainable and cost-effective interventions that could be adopted in managing PEW in CKD in resource-limited countries like Nigeria.

The limitations of the study include the fact that anorexia was not assessed using an appetite-specific questionnaire while BMI was the only anthropometric measurement taken. A large percentage of the patients had newly diagnosed CKD, therefore were probably not medically optimized. The findings may not be generalizable to non-blacks and those in high socio-economic settings. Lastly, the effect of other comorbidities such as heart failure was not taken into consideration in the analysis of factors associated with PEW in this study. However, the strength of this study is the fairly large sample size for an under-resourced single-centre setting.

## Conclusion

PEW is common in pre-dialysis CKD patients and it is associated with age, anorexia, depression, stage and duration of CKD. We strongly recommend regular assessment of PEW and depression in CKD patients, irrespective of their stage and prompt institution of management. Early intervention aimed at addressing anorexia, depression in early stages of CKD may prevent PEW and improve overall outcomes in CKD patients.

## Supporting information

**S1 Checklist. STROBE statement—checklist of items that should be included in reports of observational studies.**
(DOCX)

## Acknowledgments

The authors acknowledge the efforts of residents who helped in data collection for this study.

## Author Contributions

**Conceptualization:** Osariemen Augustine Osunbor, Enajite Ibiene Okaka, Oluseyi Ademola Adejumo.

**Data curation:** Osariemen Augustine Osunbor, Enajite Ibiene Okaka.

**Formal analysis:** Evelyn Irobere Unuigbe, Oluseyi Ademola Adejumo.

**Funding acquisition:** Osariemen Augustine Osunbor.

**Investigation:** Osariemen Augustine Osunbor, Oluseyi Ademola Adejumo.

**Methodology:** Osariemen Augustine Osunbor, Evelyn Irobere Unuigbe, Enajite Ibiene Okaka, Oluseyi Ademola Adejumo.

**Resources:** Osariemen Augustine Osunbor.

**Supervision:** Evelyn Irobere Unuigbe, Enajite Ibiene Okaka.

**Writing – original draft:** Osariemen Augustine Osunbor.

**Writing – review & editing:** Osariemen Augustine Osunbor, Evelyn Irobere Unuigbe, Enajite Ibiene Okaka, Oluseyi Ademola Adejumo.

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
