## [Decision Letter · Decision Letter 0]

17 Feb 2023

PONE-D-23-00057MALNUTRITION IN PRE-DIALYSIS CHRONIC KIDNEY DISEASE PATIENTS IN BENIN CITY, NIGERIA: A CROSS-SECTIONAL STUDYPLOS ONE

Dear Dr. ADEJUMO,

Thank you for submitting your manuscript to PLOS ONE. After careful consideration, we feel that it has merit but does not fully meet PLOS ONE’s publication criteria as it currently stands. Therefore, we invite you to submit a revised version of the manuscript that addresses the points raised during the review process.

We look forward to receiving your revised manuscript.

Kind regards,

Udeme Ekpenyong Ekrikpo, MBBS, FMCP, PhD

Academic Editor

PLOS ONE

Journal Requirements:

2. In the Methods section of your manuscript, please provide additional details regarding participant consent. In the ethics statement in the Methods and online submission information, please ensure that you have specified what type you obtained (for instance, written or verbal, and if verbal, how it was documented and witnessed). If your study included minors, state whether you obtained consent from parents or guardians. If the need for consent was waived by the ethics committee, please include this information.

  "NO- The funders had no role in study design, data collection and analysis, decision to publish, or preparation of the manuscript."

4. We noted in your submission details that a portion of your manuscript may have been presented or published elsewhere. "The abstract of this work was presented in World Congress of Nephrology in 2020 " Please clarify whether this conference proceeding or publication was peer-reviewed and formally published. If this work was previously peer-reviewed and published, in the cover letter please provide the reason that this work does not constitute dual publication and should be included in the current manuscript.

Reviewers' comments:

Reviewer's Responses to Questions

**Comments to the Author**

1. Is the manuscript technically sound, and do the data support the conclusions?

Reviewer #1: Yes

Reviewer #2: Partly

2. Has the statistical analysis been performed appropriately and rigorously? 

Reviewer #1: Yes

Reviewer #2: Yes

3. Have the authors made all data underlying the findings in their manuscript fully available?

Reviewer #1: Yes

Reviewer #2: Yes

4. Is the manuscript presented in an intelligible fashion and written in standard English?

Reviewer #1: Yes

Reviewer #2: Yes

5. Review Comments to the Author

Reviewer #1: 1- It is better to replace the word "Dialysis naive" in the methods' study design section by the word "pre-dialysis" to be the same as the title.

2- In the methods' section it is better to remove the subtitle "Sample Size Calculation" or instead the authours have to add more subtitles as "inclusion, exclusion criteria ....etc"

3- It is better to mention in inclusion criteria that they include CKD 3-5ND (not on dialysis), and remove from the exclusion criteria hemodialysis patients

Reviewer #2: My attachment includes detailed feedback to the authors. Overall the analysis is fairly straightforward and done correctly however should not be over-analyzed/ interpreted due to multiple limitations.

6. PLOS authors have the option to publish the peer review history of their article (what does this mean?). If published, this will include your full peer review and any attached files.

Reviewer #1: **Yes: **Mohammed Abdel Gawad

Reviewer #2: No

---

## [Author Response · Author response to Decision Letter 0]

24 Mar 2023

SN LINE COMMENT RESONSES

1. Line 30 Change statement: It is a predictor of adverse outcomes in CKD DONE

2. Line 31 Change statement: … malnutrition is not routinely assessed during the management of CKD in Nigeria

It is routinely assessed in well-resourced countries that have the luxury of a multi-disciplinary kidney team DONE

3. Line 49-54 Again reference your setting. For example, malnutrition is not often diagnosed early in Nigeria and worsens with progressive CKD.

Please clarify your definition of Malnutrition in your CKD setting. I don’t think low albumin (in isolation) qualifies as malnutrition. This is a problem throughout the manuscript.

Consider using the term PEW. Term coined in 2007 by the International Society of Renal Nutrition and Metabolism (ISRNM) to describe the malnutrition often seen among patients with CKD. PEW is specifically defined as the syndrome of depletion of systemic body protein and energy stores with specific diagnostic criteria (chemistry, BMI, muscle mass, dietary intake).

You can define it in your setting but it would have to comprise a combination of factors. DONE

PEW has been used to replace Malnutrition

Criteria for Diagnosis has been reviewed to be a combination of these ; 

1. Serum albumin revised to < less than 3.8g/l

2. BMI revised to < less than 23kg/m2

3. SGA 

In line with recommendation of ISRNM, we have revised the cut off values of

1. Low BMI from 18.5 to 23 kg/m2

2. Low Albumin from 3.5 to 3.8g/l

Appropriate analysis has also been done and reflected in the manuscript

4. Line 54 is redundant. Remember the obesity paradox in dialysis. sentence removed as

5. Line 57 Change underlying chronic disease condition to their underlying CKD Done

6. Line 57+58 (Consider removing lines) It is well established that malnutrition is common in low SES. This is redundant.

 Done

7. Line 62-63 I would re-word the statement to, for example: ‘ The prevalence of malnutrition is higher with more advanced CKD and into dialysis’ Done

8. Line 70-72 Please revise extensively. Malnutrition is perhaps not a direct cardiovascular risk factor but an association rather. Chronic inflammation is common in CKD dialysis patients and leads to PEW with increased protein catabolism. Also, I am not convinced that malnutrition worsens kidney disease but rather worsens associated frailty, sarcopenia and debility Done.

Suggested revision done

9. Line 73 +74 Please provide references to sub-Saharan African studies on the topic 3 references have been provided

10. Line 75 Reword the statement to include the fact that you are increasing awareness to facilitate prompt diagnosis and treatment of high-risk individuals. The aim has been rephrased

11. Line 77 Change ‘renal’ to ‘kidney’ (please use kidney throughout)

Methods:

Please clarify if you included everyone who was seen at your clinic or specifically only those who consented. I assume the former as depression scores are probably not routine clinical care. If so, how many patients were approached? Or, were some of the data collected retrospectively? If so how did you decide whom to include? 

 Done 

Clarifications

1. All those who were attending clinic and met inclusion criteria 

were included after giving their consent

2. None declined

3. Data were not collected retrospectively

12. Line 85+86 Did you include generally well patients? If not (i.e. recent hospitalization) this needs to be mentioned as a limitation as this could affect lots of variables including inflammation, and albumin (negative acute phase reactant). Apparently well CKD patients were recruited for the study

13. Line 97 + 98 Where was your control group recruited from? Control group were recruited from those working in the hospital and

 The affiliated University

14. Line 103 I am not sure why you included BMI <18.5kg/m2, This is the definition in non-CKD patients. In CKD it is <23kg/m2. The below reference is useful.

https://doi.org/10.1159/000504240

Revision done 

BMI < 23 iwas used and re-analysis done

15. Line 107 Please clarify if you mean that the same investigator performed the depression scales and SGA in all participants. Also, did you use the 7-point SGA and if so mention this. This is more reliable in the CKD population Trained research assistants were used to perform depression and SGA

 for the study participants

7-point- SGA was used. This has been included

16. Line 117 I am curious why you used this depression scale. The Beck Depression Inventory is the most validated and widely used score in CKD. Perhaps mention this as a limitation.

The below reference is useful.

Kondo, Karli1,2; Antick, Jennifer R.3,4; Ayers, Chelsea K.1; Kansagara, Devan1,5,6; Chopra, Pavan5. Depression Screening Tools for Patients with Kidney Failure: A Systematic Review. CJASN 15(12):p 1785-1795, December 2020. | DOI: 10.2215/CJN.05540420

 Yes we agree that BDI is the most commonly used screening tool.

Other screening tools have been being validated and used in CKD population

Gençöz F, Gençöz T, Soykan A. Psychometric properties of the Hamilton 

Depression Rating Scale and other physician-rated psychiatric scales for

 the assessment of depression in ESRD patients undergoing hemodialysis in

 Turkey. Psychol Health Med. 2007;12(4):450-9

17. Line 126 Please elaborate/ clarify what is meant by ethical consideration under the definition of malnutrition

Results: 

- Mention that basic demographics for both the CKD group and control appear in table 1. 

You don’t need to put everything into words only the variables of interest: age, gender, cause of CKD, and stages of CKD. 

- Were the causes of CKD biopsy proven or assumed? I am guessing the latter. Please mention this (i.e. cause as noted as the opinion of the attending Nephrologist).

- I am very surprised not to see HIV-associated CKD here. Is this included under ‘other’? This needs to be mentioned as it is a leading cause of CKD in sub-Saharan Africa and would affect the biochemical results. 

- The P value column isn’t adding value in table 1. You could say in the text that the control and CKD groups were well-matched for basic demographics (age and gender). This control group cannot be used for meaningful statistical comparison as there are too many unmeasured confounders. The control group serves just as a general idea of the prevalence of malnutrition in a ‘well’ population in your setting. This point needs to be mentioned clearly in the discussion.

- There is no p-value for the causes of CKD and malnutrition. I am surprised that there were no associations between diabetes or HIV and malnutrition. 

- HIV status and sickle cell disease are now mentioned in table 5 versus Tables 1 and 4. The periods for the duration of CKD are also not consistent between the tables.

- You need to clarify (in the methods) that the definition/ assessment of anorexia is based on SGA findings of self-reported intake and loss of appetite. Some studies use food journals and appetite-specific assessment tools. 

DONE

Discussion:

- The first paragraph needs to be reworked entirely. You can say that this study aimed to give an overall impression of the scope of malnutrition in a moderate-severe outpatient CKD cohort in Nigeria to raise awareness of the problem and intervene accordingly. For meaningful statistical comparison, you would have to consider other chronic communicable and non-communicable conditions and if these conditions are stable and managed. Similarly, the control group only gives a local snapshot of the levels of malnutrition in the general population in the general population not affected by CKD.

- You need to separate ‘previous studies’ 8-15 into lower and higher-income countries. Compare your findings to other studies in Nigeria and then to other studies in well-resourced countries.

- Mention novel aspects of your patient population. 50 is seen as young in higher-income countries. This needs to be elaborated upon and contrasted to higher-income countries where patients are much older and frailer. You wouldn’t expect to see this level of ‘malnutrition’ in this age group in well-resourced settings so this is a meaningful finding. 

- I would re-work your analysis in the 2nd paragraph using 23kg/m2 as a BMI cut-off in your setting. I see reference 22 uses a BMI of 25kg/m2. Also, mention that patients in your study are younger and would have more muscle mass than older CKD cohorts in well-resourced settings. Therefore although BMI in itself has limitations previous meta-analyses (referenced below) have shown that the risk of death in CKD stages 3–5, is reduced by 1% for every 1 kg/m2 increase in BMI. 

Ahmadi SF, Zahmatkesh G, Ahmadi E, Streja E, Rhee CM, Gillen DL, et al. Association of Body Mass Index with clinical outcomes in non-Dialysis-dependent chronic kidney disease: a systematic review and meta-analysis. Cardiorenal Med. 2015;6:37–49.

Ladhani M, Craig JC, Irving M, Clayton PA, Wong G. Obesity and the risk of cardiovascular and all-cause mortality in chronic kidney disease: a systematic review and meta-analysis. Nephrol Dial Transplant. 2017;32:439–49.

 1. ethical consideration under the definition of malnutrition

has been corrected. 

2. The result section has been revised as suggested. Other less important information removed

3. The aetiology were based on opinion of the nephrologist. 

This has been included in the method

4. In Nigeria, the incidence HIV related kidney disease has 

significantly reduced because of the success of the prevention

measures instituted nationwide. HIV is not among the 

leading four causes of CKD in Nigeria

5. We completely agree that HIV may affect the markers of nutrition

6. We did not find significant association between PEW and aetiology

of CKD

7. Matching was for gender and age. This has been included

DISCUSSION

17. Line 126 Please elaborate/ clarify what is meant by ethical consideration under the definition of malnutrition

Results: 

- Mention that basic demographics for both the CKD group and control appear in table 1. 

You don’t need to put everything into words only the variables of interest: age, gender, cause of CKD, and stages of CKD. 

- Were the causes of CKD biopsy proven or assumed? I am guessing the latter. Please mention this (i.e. cause as noted as the opinion of the attending Nephrologist).

- I am very surprised not to see HIV-associated CKD here. Is this included under ‘other’? This needs to be mentioned as it is a leading cause of CKD in sub-Saharan Africa and would affect the biochemical results. 

- The P value column isn’t adding value in table 1. You could say in the text that the control and CKD groups were well-matched for basic demographics (age and gender). This control group cannot be used for meaningful statistical comparison as there are too many unmeasured confounders. The control group serves just as a general idea of the prevalence of malnutrition in a ‘well’ population in your setting. This point needs to be mentioned clearly in the discussion.

- There is no p-value for the causes of CKD and malnutrition. I am surprised that there were no associations between diabetes or HIV and malnutrition. 

- HIV status and sickle cell disease are now mentioned in table 5 versus Tables 1 and 4. The periods for the duration of CKD are also not consistent between the tables.

- You need to clarify (in the methods) that the definition/ assessment of anorexia is based on SGA findings of self-reported intake and loss of appetite. Some studies use food journals and appetite-specific assessment tools. 

Discussion:

- The first paragraph needs to be reworked entirely. You can say that this study aimed to give an overall impression of the scope of malnutrition in a moderate-severe outpatient CKD cohort in Nigeria to raise awareness of the problem and intervene accordingly. For meaningful statistical comparison, you would have to consider other chronic communicable and non-communicable conditions and if these conditions are stable and managed. Similarly, the control group only gives a local snapshot of the levels of malnutrition in the general population in the general population not affected by CKD.

- You need to separate ‘previous studies’ 8-15 into lower and higher-income countries. Compare your findings to other studies in Nigeria and then to other studies in well-resourced countries.

- Mention novel aspects of your patient population. 50 is seen as young in higher-income countries. This needs to be elaborated upon and contrasted to higher-income countries where patients are much older and frailer. You wouldn’t expect to see this level of ‘malnutrition’ in this age group in well-resourced settings so this is a meaningful finding. 

- I would re-work your analysis in the 2nd paragraph using 23kg/m2 as a BMI cut-off in your setting. I see reference 22 uses a BMI of 25kg/m2. Also, mention that patients in your study are younger and would have more muscle mass than older CKD cohorts in well-resourced settings. Therefore although BMI in itself has limitations previous meta-analyses (referenced below) have shown that the risk of death in CKD stages 3–5, is reduced by 1% for every 1 kg/m2 increase in BMI. 

Ahmadi SF, Zahmatkesh G, Ahmadi E, Streja E, Rhee CM, Gillen DL, et al. Association of Body Mass Index with clinical outcomes in non-Dialysis-dependent chronic kidney disease: a systematic review and meta-analysis. Cardiorenal Med. 2015;6:37–49.

Ladhani M, Craig JC, Irving M, Clayton PA, Wong G. Obesity and the risk of cardiovascular and all-cause mortality in chronic kidney disease: a systematic review and meta-analysis. Nephrol Dial Transplant. 2017;32:439–49.

 1. ethical consideration under the definition of malnutrition

has been corrected. 

2. The result section has been revised as suggested. Other less important information removed

3. The aetiology were based on opinion of the nephrologist. 

This has been included in the method

4. In Nigeria, the incidence HIV related kidney disease has 

significantly reduced because of the success of the prevention

measures instituted nationwide. HIV is not among the 

leading four causes of CKD in Nigeria

5. We completely agree that HIV may affect the markers of nutrition

6. We did not find significant association between PEW and aetiology

of CKD

7. Matching was for gender and age. This has been included

8. Comparison with control downplayed in the discussion

9. We agreed with all suggestions made by the reviewing in the discussion 

Section and they have been addressed

10. The first paragraph in the discussion has been re-written

11. The discussion has been reviewed in line with the new results

12. Comparison with similar studies done in Nigeria, Africa and high-resourced

Countries have been separated and discussed individually. We have

 broken down references 8-15

13. Some peculiar findings of our study has been emphasized

14. Limitations of BMI as a marker of nutrition has been discussed

15. The implication of BMI changes with mortality has been included

16. We have included treatment modalities for improving nutrition in

CKD patients and emphasizing what could be adopted in our local

settings

Suggested references have been included

18. Paragragh 3 In paragraph 3, importantly, the association of poor nutritional state and clinical outcome may not be a cause-and-effect relationship in these studies and is overstated here. Malnutrition and low albumin may be a result of or surrogate markers of other intercurrent illnesses.

Please use kidney (line 194) instead of renal This has been done

19. 

Paragraph 4 In paragraph 4 you are providing 1 reference using a dialysis cohort. Please find references for the utility of SGA in CKD patients and how your findings compare. This merits some discussion and interpretation of your findings.

 Information about utility of SGA has been provided 

20. Paragraph 6 Paragraph 6 needs references for lines 215-218. I would mention that 44% of your patients were diagnosed within 3 months so in your setting, as you mention, prompt and timely intervention is necessary This has been done

21. Reference 9 Reference 9 is a good one (paragraph 7) but reference 10 is about appetite in dialysis cohorts (not comparable). Are there any more references in CKD patients and how did they assess appetite? I would put the pathophysiology aspects perhaps in a table or a figure and discuss appetite in CKD. Perhaps mention tools and ways to assess appetite briefly. This is more interesting than the well-established pathophysiology.

 1. Reference 10 has been removed

2. Tools to objectively assess appetite included

22. In the discussion, there is no mention of interventions for malnutrition in the outpatient CKD setting. Some interventions have been successful such as ONS (see below reference). There is also renewed interest in physical activity for improving overall nutritional status in CKD. You can mention/ call for the need for cost-effective interventions in your setting where ONS is not perhaps freely available. 

https://doi.org/10.1177/20543581211069008

Interventions to treat or prevent PEW have been included

23. You can mention strengths, a fairly large sample size for an under-resourced single-centre setting with apparently no missing data. There are many limitations:

- If you used the original 3-point versus 7-point SGA scale this is a limitation.

- As far as I know, the diagnostic accuracy of the Hamilton depression score has not been reliably proven in CKD.

- You did not include other co-morbidities such as heart failure in your analysis and whether these were well managed.

- Anorexia was only assessed from the SGA cross-sectionally, not an appetite-specific questionnaire

- BMI is the only anthropometry- no muscle measurements/body composition analysis

- Screening and assessments were performed once only and it is, therefore, unclear whether there were significant changes in variables during the follow-up period. A large percentage of your patients had newly diagnosed CKD therefore are probably not medically optimized.

- I assume your population was mostly Black- therefore your findings may not be generalizable to other low SES settings.

 Suggestions about limitations and strength of study were accepted and

 reflected

8. Comparison with control downplayed in the discussion

9. We agreed with all suggestions made by the reviewing in the discussion 

Section and they have been addressed

10. The first paragraph in the discussion has been re-written

11. The discussion has been reviewed in line with the new results

12. Comparison with similar studies done in Nigeria, Africa and high-resourced

Countries have been separated and discussed individually. We have

 broken down references 8-15

13. Some peculiar findings of our study has been emphasized

14. Limitations of BMI as a marker of nutrition has been discussed

15. The implication of BMI changes with mortality has been included

16. We have included treatment modalities for improving nutrition in

CKD patients and emphasizing what could be adopted in our local

settings

Suggested references have been included

18. Paragragh 3 In paragraph 3, importantly, the association of poor nutritional state and clinical outcome may not be a cause-and-effect relationship in these studies and is overstated here. Malnutrition and low albumin may be a result of or surrogate markers of other intercurrent illnesses.

Please use kidney (line 194) instead of renal This has been done

19. 

Paragraph 4 In paragraph 4 you are providing 1 reference using a dialysis cohort. Please find references for the utility of SGA in CKD patients and how your findings compare. This merits some discussion and interpretation of your findings.

 Information about utility of SGA has been provided 

20. Paragraph 6 Paragraph 6 needs references for lines 215-218. I would mention that 44% of your patients were diagnosed within 3 months so in your setting, as you mention, prompt and timely intervention is necessary This has been done

21. Reference 9 Reference 9 is a good one (paragraph 7) but reference 10 is about appetite in dialysis cohorts (not comparable). Are there any more references in CKD patients and how did they assess appetite? I would put the pathophysiology aspects perhaps in a table or a figure and discuss appetite in CKD. Perhaps mention tools and ways to assess appetite briefly. This is more interesting than the well-established pathophysiology.

 1. Reference 10 has been removed

2. Tools to objectively assess appetite included

22. In the discussion, there is no mention of interventions for malnutrition in the outpatient CKD setting. Some interventions have been successful such as ONS (see below reference). There is also renewed interest in physical activity for improving overall nutritional status in CKD. You can mention/ call for the need for cost-effective interventions in your setting where ONS is not perhaps freely available. 

https://doi.org/10.1177/20543581211069008

Interventions to treat or prevent PEW have been included

23. You can mention strengths, a fairly large sample size for an under-resourced single-centre setting with apparently no missing data. There are many limitations:

- If you used the original 3-point versus 7-point SGA scale this is a limitation.

- As far as I know, the diagnostic accuracy of the Hamilton depression score has not been reliably proven in CKD.

- You did not include other co-morbidities such as heart failure in your analysis and whether these were well managed.

- Anorexia was only assessed from the SGA cross-sectionally, not an appetite-specific questionnaire

- BMI is the only anthropometry- no muscle measurements/body composition analysis

- Screening and assessments were performed once only and it is, therefore, unclear whether there were significant changes in variables during the follow-up period. A large percentage of your patients had newly diagnosed CKD therefore are probably not medically optimized.

- I assume your population was mostly Black- therefore your findings may not be generalizable to other low SES settings.

 Suggestions about limitations and strength of study were accepted and

 reflected

---

## [Editor Report · Decision Letter 1]

6 Apr 2023

PONE-D-23-00057R1PROTEIN ENERGY WASTING IN PRE-DIALYSIS CHRONIC KIDNEY DISEASE PATIENTS IN BENIN CITY, NIGERIA: A CROSS-SECTIONAL STUDYPLOS ONE

Dear Dr. ADEJUMO,

Thank you for submitting your manuscript to PLOS ONE. After careful consideration, we feel that it has merit but does not fully meet PLOS ONE’s publication criteria as it currently stands. Therefore, we invite you to submit a revised version of the manuscript that addresses the points raised during the review process.

ACADEMIC EDITOR:

Kindly correct the 95% confidence interval in the abstract. "On multiple logistic regression, the factors associated with PEW in CKD were

middle aged (adjusted odds ratio: 12.50; confidence interval: 3.42-4.50; p <0.001).........." and re-submit.

We look forward to receiving your revised manuscript.

Kind regards,

Udeme Ekpenyong Ekrikpo, MBBS, FMCP, PhD

Academic Editor

PLOS ONE
---

## [Author Response · Author response to Decision Letter 1]

15 Apr 2023

Comment: Kindly correct the 95% confidence interval in the abstract. "On multiple logistic regression, the factors associated with PEW in CKD were

middle aged (adjusted odds ratio: 12.50; confidence interval: 3.42-4.50; p <0.001).........." and re-submit.

Response: This been corrected

---

## [Editor Report · Decision Letter 2]

9 May 2023

PROTEIN ENERGY WASTING IN PRE-DIALYSIS CHRONIC KIDNEY DISEASE PATIENTS IN BENIN CITY, NIGERIA: A CROSS-SECTIONAL STUDY

PONE-D-23-00057R2

Dear Dr Adejumo

We’re pleased to inform you that your manuscript has been judged scientifically suitable for publication and will be formally accepted for publication once it meets all outstanding technical requirements.

Kind regards,

Udeme Ekpenyong Ekrikpo, MBBS, FMCP, PhD

Academic Editor

PLOS ONE
---

## [Editor Report · Acceptance letter]

15 May 2023

PONE-D-23-00057R2 

PROTEIN ENERGY WASTING IN PRE-DIALYSIS CHRONIC KIDNEY DISEASE PATIENTS IN BENIN CITY, NIGERIA: A CROSS-SECTIONAL STUDY 

Dear Dr. Adejumo:

I'm pleased to inform you that your manuscript has been deemed suitable for publication in PLOS ONE. Congratulations! Your manuscript is now with our production department. 

Kind regards, 

on behalf of

Dr. Udeme Ekpenyong Ekrikpo 

Academic Editor

PLOS ONE